# Myeloid HIF1α Is Involved in the Extent of Orthodontically Induced Tooth Movement

**DOI:** 10.3390/biomedicines9070796

**Published:** 2021-07-08

**Authors:** Christian Kirschneck, Nadine Straßmair, Fabian Cieplik, Eva Paddenberg, Jonathan Jantsch, Peter Proff, Agnes Schröder

**Affiliations:** 1Department of Orthodontics, University Medical Centre of Regensburg, D-93053 Regensburg, Germany; nadine.strassmair@stud.uni-regensburg.de (N.S.); eva.paddenberg@ukr.de (E.P.); peter.proff@ukr.de (P.P.); agnes.schroeder@ukr.de (A.S.); 2Department of Operative Dentistry and Periodontology, University Medical Centre of Regensburg, D-93053 Regensburg, Germany; fabian.cieplik@ukr.de; 3Institute of Microbiology and Hygiene, University Medical Centre of Regensburg, D-93053 Regensburg, Germany; jonathan.jantsch@ukr.de

**Keywords:** orthodontic tooth movement, HIF1α, macrophage, µCT

## Abstract

During orthodontic tooth movement, transcription factor hypoxia-inducible factor 1α (HIF1α) is stabilised in the periodontal ligament. While HIF1α in periodontal ligament fibroblasts can be stabilised by mechanical compression, in macrophages pressure application alone is not sufficient to stabilise HIF1α. The present study was conducted to investigate the role of myeloid HIF1α during orthodontic tooth movement. Orthodontic tooth movement was performed in wildtype and *Hif1α*^Δmyel^ mice lacking HIF1α expression in myeloid cells. Subsequently, µCT images were obtained to determine periodontal bone loss, extent of orthodontic tooth movement and bone density. RNA was isolated from the periodontal ligament of the control side and the orthodontically treated side, and the expression of genes involved in bone remodelling was investigated. The extent of tooth movement was increased in *Hif1α*^Δmyel^ mice. This may be due to the lower bone density of the *Hif1α*^Δmyel^ mice. Deletion of myeloid Hif1α was associated with increased expression of *Ctsk* and *Acp5*, while both *Rankl* and its decoy receptor *Opg* were increased. HIF1α from myeloid cells thus appears to play a regulatory role in orthodontic tooth movement.

## 1. Introduction

Orthodontic tooth movement is therapeutically induced to correct malpositioned and displaced teeth to improve oral function, occlusion and aesthetics. This is achieved by applying mechanical forces via fixed and removable orthodontic appliances, creating pressure and tension zones within the periodontal ligament, which is a fibrous connective tissue linking teeth with their surrounding alveolar bone. This induces bone resorption by osteoclasts and bone deposition by osteoblasts depending on the magnitude, direction and duration of the applied force [1,2]. The mechanical forces transmitted to periodontal fibroblasts [3], macrophages [4], T cells [5] and other cell populations within the periodontal ligament induce a sterile inflammatory process favouring the increased expression of inflammatory regulators such as interleukins and prostaglandins and of signalling molecules controlling bone turnover such as receptor activator of NF-kB ligand (RANKL) [3,6,7,8]. RANKL binds to the receptor activator of NF-kB (RANK) receptor, which is expressed on the surface of osteoclast progenitor cells [9]. RANKL/RANK signalling regulates osteoclast formation, activation and survival in normal bone remodelling, as well as in a variety of pathological conditions characterised by increased bone turnover [9], including periodontitis—a recent systematic review and meta-analysis revealed that RANKL mRNA levels were significantly higher in gingival tissue as well as protein levels in the gingival crevicular fluid (GCF) of individuals suffering from periodontitis [10]. Osteoprotegerin (OPG), which functions as a decoy receptor of RANKL and can be released from osteoblasts as well as periodontal ligament (PDL) fibroblasts, protects bone from excessive resorption by binding to RANKL and preventing its binding to the membrane-bound RANK receptor on osteoclast precursor cells and thus their differentiation and fusion to active bone-resorbing osteoclasts [9]. In this regard, it is not the absolute expression of RANKL that is decisive for osteoclast activity, but rather the ratio of RANKL to OPG expression, both for orthodontic tooth movement [2], which requires controlled osteoclastogenesis and osteoclast activity in the direction of tooth movement, as well as periodontitis [10,11]. A recent systematic review reports that (pre)clinical evidence regarding periapical as well as periodontal lesions hints at an increase of the RANKL/OPG ratio as a primary determinant of osteolytic activity in the etiopathology of periodontitis, whereas a decreased RANKL/OPG ratio seems to be associated with inactive lesions [11]. Thus, the RANKL/OPG system plays a crucial role both in orthodontic tooth movement as well as in the etiopathology of periodontitis [11,12]. Apart from the RANKL/OPG system, a distinct remodelling of the extracellular matrix (ECM) of the PDL is also required for OTM, which seems to be primarily controlled by different matrix metalloproteinases (MMPs) and their tissue inhibitors (TIMPs)—both released by PDL fibroblasts upon force application [13]. These and various other molecules have been suggested as possible biomarkers in saliva or gingival crevicular fluid for monitoring or predicting OTM and possible side effects such as root resorptions. Allen et al. identified 20 OTM-related biomarkers in saliva in their systematic review [14] and Alhadlaq 34 OTM-related biomarkers in GCF [15] with some biomarkers seeming to occur both in saliva and GCF (e.g., RANKL/OPG, leptin) and others not (e.g., apoE, CLU and CRISP-3).

Next to the onset of a sterile inflammation within the periodontal ligament induced by the mechanical forces applied during orthodontic treatment, a compression of blood vessels occurs leading to a locally reduced oxygen supply [1]. In addition to periodontal fibroblasts, which comprise the main cell population of the periodontal ligament, immune cells such as macrophages are exposed to mechanical stress and the concomitantly reduced oxygen supply [4,16,17]. Macrophages have been shown to modulate the differentiation of osteoclast progenitor cells into bone-resorbing osteoclasts and thus promote tooth movement [16] by secreting a variety of cytokines such as tumour necrosis factor (TNF) or interleukin 6 (IL-6), which stimulate bone resorption by increasing the expression of RANKL [4]. To adapt to a reduced oxygen supply, angiogenic factors are increasingly expressed by periodontal ligament cells so that an adequate blood supply associated with an improved oxygen saturation can be restored [1]. During hypoxia, hypoxia-inducible factor 1α (HIF1α) is stabilised in tissues, playing a crucial mediating role in tissue adaptation to hypoxic conditions through the transcriptional activation of more than a hundred different target genes that regulate vital biological processes [18,19]. To counteract hypoxia, HIF1α activates genes for enhanced oxygen transport, angiogenesis, vasodilation and anaerobic glycolysis. In this process, HIF1α binds to matching elements in the promoter or enhancer region and thereby activates the transcription of these genes [18,19]. In a study by Wan et al. (2008), it was shown that HIF1α is significantly involved in bone healing [20]. The results of this study identify the HIF1α signalling pathway as a critical mediator of the neoangiogenesis required for skeletal regeneration and suggest the use of HIF activators as a potential therapy to enhance bone healing [20]. Thus, it can be surmised that HIF1α plays an important role during orthodontic tooth movement.

As orthodontic tooth movement (OTM) is a multicellular process, cell culture experiments are only suitable to a limited extent for exploring possible effectors of orthodontically induced tooth movement. It was recently demonstrated in a study by Schröder et al. (2020) that in mechanically stressed macrophages HIF1α is stabilised by the concomitantly reduced oxygen supply associated with orthodontic therapy rather than by mechanical strain itself [17]. The opposite appears to be true in periodontal ligament fibroblasts with HIF1α stabilisation induced by mechanical strain rather than oxygen deprivation [21]. These results suggest that both cell types react differently to hypoxic conditions and mechanical compression occurring simultaneously during orthodontic treatment. Based on these in vitro results, this in vivo study was conducted to clarify the role of HIF1α in myeloid cells during orthodontic treatment.

## 2. Materials and Methods

### 2.1. Animal Experiment

The animal experiments were conducted in compliance with the German Animal Protection Act and with approval of the Government of Lower Franconia, Bavaria, Germany (approval ID: 55.2-2532-2-567, 25th January 2018). The termination criteria were defined, and mice gross weight was controlled daily to avoid unnecessary suffering. The mice were kept in a conventional animal laboratory with a constant temperature of 21 °C and a 12 h light–dark rhythm with the light phase between 07:00 and 19:00. The mice had free access to water and a standard diet (V1535, ssniff, Soest, Germany) with food pellets softened with water after the insertion of the orthodontic appliance. The mice were anaesthetised with xylazine/ketamine (1:3) diluted 1:5 with saline. A total of 20 LysM^WT^*Hif1α*^fl/fl^ (control) and 20 LysM^Cre^*Hif1α*^fl/fl^ (*Hif1α*^Δmyel^) were examined. LysM^Cre^*Hif1α*^fl/fl^ mice do not express HIF1α in myeloid cells such as macrophages. Eight-week-old males were used for the experiments. An experienced scientist inserted an elastic band (diameter 0.3 mm; Inwaria Stores GmbH, Trier, Germany) between the first and second molars of the left upper jaw under anaesthesia using two Mosquito clamps (straight, with teeth) and then shortened accordingly after pre-expansion with an orthodontic auxiliary wire (Ø 0.08 mm), according to an established and validated protocol [22,23]. The diameter of 0.3 mm was the largest diameter insertable in the interdental space leading to a reciprocal movement of the first molar in the anterior and the second molar in the posterior direction upon relaxation of the compressed band [22]. The contralateral right jaw side was left untreated and served as the control side. The condition of the animals was monitored and documented daily by experienced scientists. Seven days after the insertion of the elastic band, the animals were euthanised. For the µCT analysis, ten samples per genotype were fixed in 5% formalin solution for 24 h. The formalin solution was then diluted to 0.1% with PBS. For the RNA analysis, ten maxillary samples per genotype were kept in liquid nitrogen immediately after removal and stored at −80 °C until further use.

### 2.2. RNA Analysis

The jaws were dissected on a cooling plate. The first molar was retrieved with the surrounding periodontal tissue and transferred to a pre-cooled cylinder as described before [24]. This was then pulverised with a Retsch mill (frequency 25/s, 2 × 30 s, MM200). PureLink RNA Mini Kit (12183018, Thermo Fischer Scientific Inc., Dreieich, Germany) was used for RNA isolation, according to the manufacturer’s instructions. Mouse biopsies were removed from the freezer and temporarily stored in liquid nitrogen. The isolated RNA was then measured photometrically (NanoPhotometer, Implen GmbH, Munich, Germany). For the cDNA synthesis, an equal amount of RNA was mixed with a master-mix consisting of 2 µL MMLV buffer (M531A, Promega, Walldorf, Germany), 0.5 µL OligodT (SO132, Thermo Fisher Scientific Inc., Dreieich, Germany), 0.5 µL Random Hexamers (SO142, Thermo Fisher Scientific Inc., Dreieich, Germany), 0.5 µL 10 mM dNTPs (L785.1/2, Carl Roth, Karlsruhe, Germany), 0.5 µL RNase inhibitor (EO0381, Thermo Fisher Scientific Inc., Dreieich, Germany) and 0.5 µL reverse transcriptase (M170B, Promega, Walldorf, Germany) per sample. The samples were incubated for 1 h at 37 °C. The reverse transcriptase was inactivated by heating at 90 °C for 2 min. The cDNA was diluted 1:5. A separate qPCR primer mix was prepared for each gene to be examined, containing 0.25 µL forward primer, 0.25 µL reverse primer (Table 1), 5 µL Luna Universal qPCR Mix (M3003E, New England BioLabs, Frankfurt am Main, Germany) and 3 µL RNase-free H_2_O_dd_ (T143, Carl Roth). The primers used were specific for the genes to be investigated (Table 1). To determine the relative gene expression, the formula 2^−ΔCq^ was used [24,25] with ΔC_q_ = C_q_ (target gene) − C_q_ (geometric mean *Eef1a1/Ywhaz*).

### 2.3. MicroCT Analysis

To investigate orthodontic tooth movement, periodontal bone loss and bone density, µCT imaging (Phoenix vltomelxs 240/180, GE Healthcare, Solingen, Germany) was implemented. The programme VGSTUDIO MAX 3.2.4 (Volume Graphics GmbH, Heidelberg, Germany) was used for evaluation. Jaws were aligned in the sagittal plane at the mesial root of the first molar, in the horizontal plane using the occlusal plane and in the vertical plane using the palatal suture as described before [26]. The smallest approximal distance between the first and second molars, periodontal bone loss and bone density were determined. All measurements were made both at the orthodontically treated (OTM) jaw side and on the control side. Periodontal bone loss was determined in the coronal plane using a vernier calliper (Abb. 4a). The reference points were the cemento-enamel junction and the upper edge of the alveolar bone. To measure the approximal distance between the first and second molars, the smallest distance between the convex sides of the enamel of the two teeth was determined in the sagittal plane. This was measured with a digital calliper. In order to determine bone density, a region of interest (ROI, 0.3 × 0.3 × 0.3 mm height × width × depth) was set interradicularly at the first molar. The region of interest was placed to not encompass the tooth roots nor the periodontal gap.

### 2.4. Data Analysis and Statistics

The statistical evaluation was carried out with GraphPad Prism Version 9.0 (GraphPad software). Normal distribution of data was tested using the Shapiro–Wilk test. Depending on the data distribution, either an ANOVA followed by Fisher’s LSD tests or a Welch-corrected ANOVA followed by unpaired t tests was performed. If *p* < 0.05, the differences were rated as statistically significant. Each symbol corresponds to a data point. The horizontal lines represent the mean and the vertical lines the standard error of the mean.

## 3. Results

### 3.1. Effects of Myeloid Hif1α Deletion and Orthodontic Treatment on Genes Involved in Bone Formation

First, the gene expression of bone-forming parameters such as alkaline phosphatase (*Alp*) and Runt-related transcription factor 2 (*Runx2*) was investigated. In the case of *Alp*, a significant increase in gene expression was observed in animals lacking Hif1α in myeloid cells after orthodontic treatment (*p* < 0.001; Figure 1a). In wildtypes, only a trend towards an increase in the gene expression of *Alp* was detected, when comparing control and orthodontically treated jaw sides (*p* = 0.055). Neither at the control (*p* = 0.77) nor at the orthodontically treated side (*p* = 0.176) could a significant difference between the different genotypes be determined. *Runx2* did not show significantly increased expression in either wildtype (*p* = 0.201) or *Hif1α*^Δmyel^ mice (*p* = 0.152) after orthodontic treatment. There were also no significant differences between the two genotypes, neither at the control (*p* = 0.529) nor at the OTM side (*p* = 0.523; Figure 1b).

### 3.2. Effects of Myeloid Hif1α Deletion and Orthodontic Treatment on Genes Involved in Bone Modulation

Subsequently, the gene expression of bone-modulating genes such as prostaglandin synthase 2 (*Ptgs2*; Figure 2a), osteoprotegerin (*Opg*; Figure 2b) and RANK ligand (receptor activator of NF-κB-ligand, *Rankl*; Figure 2c), as well as the *Rankl/Opg* ratio, (Figure 2d) was examined. *Ptgs2* showed significant upregulation in both wildtype (*p* = 0.002) and *Hif1α*^Δmyel^ mice (*p* < 0.001), when orthodontic tooth movement was induced. However, no significant difference in *Ptgs2* gene expression could be detected between the different genotypes, neither at the control side (*p =* 0.145) nor at the OTM side (*p =* 0.093; Figure 2a). For *Opg*, there was only a significant upregulation in the *Hif1α*^Δmyel^ mice between the control and the OTM side (*p* < 0.001). No significant upregulation was detected between untreated and treated jaw sides in wildtype animals (*p =* 0.32). In addition, no significant differences were found at the control side (*p =* 0.264) or the orthodontically treated side (*p =* 0.077) between the different genotypes (Figure 2b). *Rankl* gene expression was elevated in both wildtype (*p =* 0.004) and *Hif1α*^Δmyel^ mice (*p <* 0.001) when the first molar was orthodontically moved (Figure 2c). Surprisingly, a significant *Rankl* difference between the two genotypes was also found for the OTM sides (*p <* 0.001), while no significant difference was detected for the control sides (*p =* 0.93). Finally, looking at the *Rankl/Opg* ratio, we observed an upregulation in both wildtype mice (*p =* 0.022) and mice without *Hif1α* in myeloid cells (*p =* 0.005) when OTM was initiated (Figure 2d). However, no significant differences could be detected at the control (*p =* 0.264) or the OTM (*p =* 0.098) jaw sides between the two genotypes.

### 3.3. Effects of Myeloid Hif1α Deletion and Orthodontic Treatment on Genes Involved in Bone Resorption

Next, we examined the expression of osteoclast-specific genes, which are involved in bone degradation. Orthodontic treatment increased the gene expression of Cathepsin K (*Ctsk*) in both wildtype mice (*p =* 0.005) and mice without *Hif1α* in myeloid cells (*p <* 0.001; Figure 3a). There was no significant difference between the two genotypes either at the control side (*p =* 0.967) or at the OTM side (*p =* 0.182). With acid phosphatase 5 (*Acp5*), the same effect was observed after orthodontic tooth movement (Figure 3b). Accordingly, a significant increase in gene expression was detected both in wildtype (*p <* 0.001) and *Hif1α*^Δmyel^ mice (*p <* 0.001) at the OTM jaw side compared to the control side. Surprisingly, a significantly increased gene expression was also detected at the OTM side in *Hif1α*^Δmyel^ mice compared to controls (*p =* 0.003), while no significant difference between the two genotypes was found at the control side (*p =* 0.582; Figure 3b). 

### 3.4. Effects of Myeloid Hif1α Deletion and Orthodontic Treatment on Periodontal Bone Loss and Orthodontic Tooth Movement

Next, we determined periodontal bone loss at the control and orthodontically treated jaw sides. We observed a significant increase in bone resorption in wildtype mice when orthodontic tooth movement occurred (*p =* 0.003; Figure 4a). The same effect was detected in *Hif1α*^Δmyel^ mice (*p <* 0.001). No significant difference could be found between the different genotypes at the control side (*p =* 0.125) or the OTM side (*p =* 0.124). To determine the extent of orthodontic tooth movement, we measured the approximal distance between the first and second molars in both groups (Figure 4b). As expected, no tooth movement was found at the untreated control side. There was also no significant difference between the two genotypes at the control side (*p* > 0.999). In both wildtype (*p <* 0.001) and *Hif1α*^Δmyel^ mice (*p <* 0.001), a statistically significant tooth movement was induced at the orthodontically treated side compared to the untreated control side (Figure 4b). In addition, we detected increased tooth movement in *Hif1α*^Δmyel^ mice compared to the wildtype animals (*p =* 0.001; Figure 4b). Finally, the bone density of the alveolar bone at the first upper molar was examined (Figure 4c). As expected, orthodontic tooth movement significantly decreased bone density compared to the control jaw side both in wildtype (*p =* 0.002) and *Hif1α*^Δmyel^ mice (*p <* 0.001). Surprisingly, a significant difference was also found at the OTM side between wildtype and *Hif1α*^Δmyel^ mice (*p =* 0.021). At the untreated control side, no significant difference could be detected between the two genotypes (*p =* 0.982; Figure 4c).

## 4. Discussion

As expected, orthodontic tooth movement increased the expression of genes involved in bone formation and resorption [22,23]. Tooth movement was induced in both genotypes. Furthermore, we observed decreased bone density and by tendency increased periodontal bone loss after orthodontic treatment in wildtype and *Hif1α*^Δmyel^ mice. Deletion of HIF1α in myeloid cells had no impact on the expression of genes involved in bone remodelling or on bone density without orthodontic treatment. However, *Ptgs2*, *Rankl*, *Acp5* and by tendency *Ctsk* gene expression were further elevated in mice without myeloid HIF1α after orthodontic treatment compared to wildtype mice. This was accompanied by reduced bone density and resulted in an accelerated tooth movement in mice with HIF1α deletion, indicating a bone-protective role of HIF1α in myeloid cells during orthodontic treatment.

The classical function of HIF1α is to control cell response to hypoxic environmental conditions [18,19]. HIF1α regulates the expression of genes involved in new blood vessel formation and dilation to improve tissue oxygenation through transcriptional control. During orthodontic tooth movement, compression of blood vessels results in a reduced oxygen supply [1,27,28]. This is associated with elevated HIF1α stabilisation in the periodontal ligament. In addition to periodontal ligament fibroblasts, immune cells such as macrophages are also found in the periodontal ligament and are exposed to mechanical stress [16]. Both cell types, i.e., periodontal ligament fibroblasts and macrophages, react with HIF1α stabilisation in an in vitro model of orthodontic tooth movement [4,21]. However, in fibroblasts, this is more likely to be caused by the mechanical stress itself [21], whereas in macrophages it is rather caused by the concomitantly reduced oxygen content [17].

HIF1α is involved in immune response regulation, as infected and inflamed tissues are associated with a reduced oxygen supply. HIF1α is widely expressed in all immune cells including myeloid cells such as macrophages and dendritic cells [29,30]. Of note, HIF1α deletion in myeloid cells does not affect their differentiation but impairs the invasion and motility of macrophages as response to infections [29,31]. Since orthodontic tooth movement induces sterile inflammation in the periodontal ligament, myeloid HIF1α deletion may also impair the invasion of macrophages to the periodontal ligament, thereby affecting the secretion of various cytokines and chemokines by macrophages during mechanical strain and under hypoxic conditions [4,17]. It has been found that compression leads to an induction of inflammatory factors in macrophages in vitro [4]. HIF1α seems to be involved in the transcriptional regulation of prostaglandin synthase 2 (*Ptgs2*), thereby controlling prostaglandin E2 release. However, we detected no differences in *Ptgs2* gene expression between wildtype and *Hif1α*^Δmyel^ mice without and with orthodontic treatment. This could be due to a possible compensatory role of periodontal ligament fibroblasts during orthodontic tooth movement, which react to compressive force with increased *Ptgs2* gene expression [7,21]. The by tendency observed increase of *Ptgs2* gene expression in *Hif1α*^Δmyel^ mice during OTM, however, would be in accordance with previous research, showing that *Ptgs2* expression is regulated by *Hif1α* [32,33]. In macrophages, on the other hand, an uncoupling of compressive force and oxygen supply in vitro reduced HIF1α stabilisation without affecting *Ptgs2* gene expression or prostaglandin E2 release [17].

Apart from controlling the response of cells to hypoxia and regulating immune responses, it is known that HIF1α is involved in skeletal development [34] and the control of bone resorption activity [35]. We detected impaired *Rankl* gene expression at the orthodontically treated jaw side in *Hif1α*^Δmyel^ mice and by tendency also a reduced *Rankl/Opg* expression ratio, which could be due to impaired macrophage invasion or the effects of myeloid *Hif1α* deletion on periodontal ligament fibroblasts. RANKL regulates the differentiation of osteoclast precursor cells to bone-resorbing osteoclasts by binding to the RANK receptor on osteoclast precursor cells, initiating their differentiation and fusion to bone-resorbing osteoclasts [9,12]. Therefore, increased RANKL expression is associated with increased bone resorption [9]. This would explain the observed increase in orthodontic tooth movement velocity as well as the reduction of bone density and by tendency of the periodontal bone level. In our study, elevated *Rankl* gene expression was associated with increased *Acp5* and by tendency *Ctsk* gene expression indicating enhanced osteoclast activity. *Acp5*-coded tartrate-resistant acid phosphatase (TRAP) is a known marker for osteoclast activity [36] and *Ctsk*-coded Cathepsin K is an osteoclast-secreted protease able to cleave telopeptides and the triple helix of type I collagen fibres as well as matrix-metalloproteinase-9 (MMP-9) thus activating MMP-9 [37], with both enzymes enabling the degradation of the organic components of bone during bone resorption [2]. Studies on the role of HIF1α in bone development and remodelling demonstrated that HIF1α induction led to reduced osteoclastogenesis but increased resorptive activity of mature osteoclasts [35]. In mice, HIF1α deletion is associated with a decrease in trabecular bone volume, as HIF1α is involved in osteoclast activation [38,39]. In contrast, the overexpression of HIF1α has been shown to reduce osteoblast differentiation, with osteoclastogenesis shown to be elevated [40]. Initially, we detected no differences in alveolar bone density between wildtype and *Hif1α*^Δmyel^ mice. After orthodontic treatment, however, mice lacking HIF1α in myeloid cells had a reduced bone density and accelerated tooth movement compared to wildtype animals, indicating a bone-protective activity of myeloid HIF1α during orthodontic tooth movement, which is corroborated by our findings on increased osteoclast activity and *Rankl* expression under myeloid *Hif1α* deletion as well as by the not significantly altered *Alp* and *Runx2* expression in *Hif1α*^Δmyel^ mice during OTM, as *Alp* and *Runx2* are marker genes for osteoblast activity [41].

## 5. Conclusions

Deletion of myeloid HIF1α impaired *Rankl* and *Acp5* gene expression led to reduced bone density and increased orthodontically induced tooth movement. Therefore, this study demonstrates that myeloid HIF1α seems to have a bone-protective role during orthodontic tooth movement. A targeted stabilisation of myeloid HIF1α during orthodontic treatment, as was done experimentally via DMOG (dimethyloxallyl glycine) [32], might thus in the future be a possible therapeutic approach to reduce treatment-related periodontal hazards, which needs to be investigated in further studies.

## Figures and Tables

**Figure 1 biomedicines-09-00796-f001:**
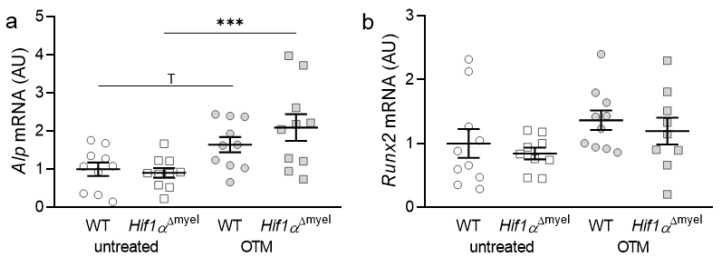
Gene expression of *Alp* (**a**) and *Runx2* (**b**) at the orthodontically treated (OTM) and the untreated control jaw side in wildtype mice (WT) and mice without HIF1α in myeloid cells (*Hif1α*^Δmyel^). Statistics: ANOVA with uncorrected Fisher’s LSD post-hoc tests (*Alp*) or Welch-corrected ANOVA with unpaired t tests (*Runx2*). *n* ≥ 9; *** *p* ≤ 0.001, ^T^
*p* ≤ 0.1.

**Figure 2 biomedicines-09-00796-f002:**
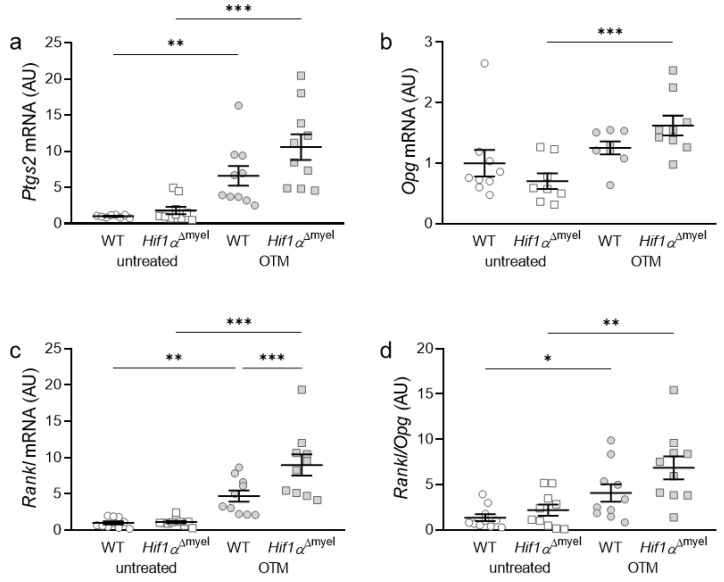
Gene expression of *Ptgs2* (**a**), *Opg* (**b**) and *Rankl* (**c**) as well as the *Rankl/Opg* ratio (**d**) at the orthodontically treated (OTM) and the untreated control jaw side in wildtype mice (WT) and mice without HIF1α in myeloid cells (*Hif1α*^Δmyel^). Statistics: ANOVA with uncorrected Fisher’s LSD post-hoc tests (*Rankl*) or Welch-corrected ANOVA with unpaired t tests (*Ptgs2, Opg, Rankl/Opg*). *n* ≥ 8; * *p <* 0.05; ** *p <* 0.01; *** *p*
*<* 0.001.

**Figure 3 biomedicines-09-00796-f003:**
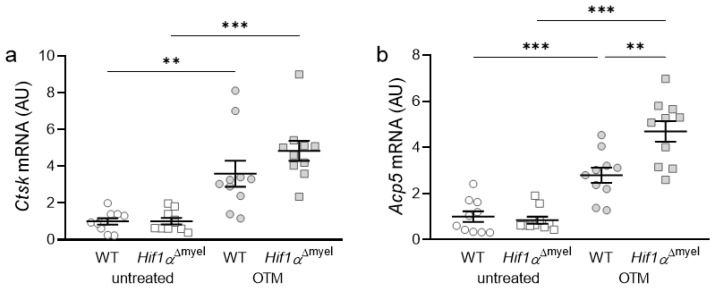
Gene expression of *Ctsk* (**a**) and *Acp5* (**b**) at the orthodontically treated (OTM) and the untreated control jaw side in wildtype mice (WT) and mice without HIF1α in myeloid cells (*Hif1α*^Δmyel^). Statistics: Welch-corrected ANOVA with unpaired *t* tests. *n* = 10; ** *p <* 0.01; *** *p <* 0.001.

**Figure 4 biomedicines-09-00796-f004:**
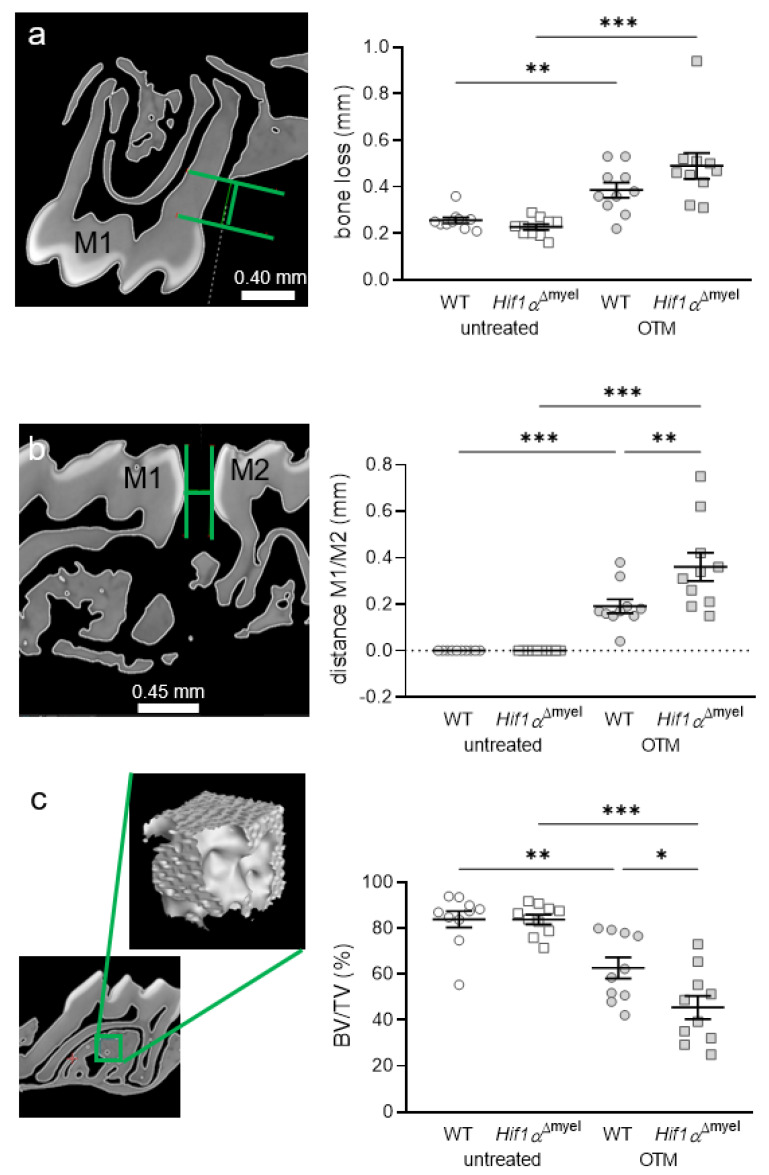
Periodontal bone loss (**a**) and orthodontic tooth movement (**b**) as the distance between the first (M1) and second (M2) molars, as well as bone density (**c**), at the orthodontically treated (OTM) and the untreated control jaw side in wildtype mice (WT) and mice without HIF1α in myeloid cells (*Hif1α*^Δmyel^). Statistics: ANOVA with uncorrected Fisher’s LSD post-hoc tests (**b**) or Welch-corrected ANOVA with unpaired *t* tests (a,c). *n* = 10; * *p <* 0.05; ** *p <* 0.01; *** *p <* 0.001.

**Table 1 biomedicines-09-00796-t001:** Primers for reference (*Eef1a1/Ywhaz*) and target genes used in RT-qPCR.

Gene	Gene Name	5′-Forward Primer-3′	5′-Reverse Primer-3′
*Acp5*	Acid Phosphatase 5, Tartrate Resistant	ATACGGGGTCACTGCCTACC	TCGTTGATGTCGCACAGAGG
*Alp*	Alkaline Phosphatase	GGGTACAAGGCTAGATGGC	AGTTCAGTGCGGTTCCAGAC
*Ctsk*	Cathepsin K	GACCCATCTCTGTGTCCATCG	CCATAGCCCACCACCAACAC
*Eef1a1*	Eukaryotic Translation Elongation Factor 1 Alpha 1	AAAACATGATTACAGGCACATCCC	GCCCGTTCTTGGAGATACCAG
*Opg*	Osteoprotegerin	CCTTGCCCTGACCACTCTTAT	CACACACTCGGTTGTGGGT
*Ptgs2*	Prostaglandin-Endoperoxide Synthase 2	TCCCTGAAGCCGTACACATC	TCCCCAAAGATAGCATCTGGAC
*Rankl*	Receptor Activator of NF-κB Ligand	AAACGCAGATTTGCAGGACTC	CCCCACAATGTGTTGCAGTTC
*Runx2*	Runt-related Transcription Factor 2	CTCCCTGAACTCTGCACCAAG	GAGTGGATGGATGGGGATGTC
*Ywhaz*	Tryptophan 5-Monooxygenase Activation Protein Zeta	AATGCTTCGCAACCAGAAAGC	TGGTATGCTTGCTGTGACTGG

## Data Availability

All datasets are publicly available either as supplementary information to this article or upon request from the corresponding author.

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
