# Peer review of "Myeloid HIF1α Is Involved in the Extent of Orthodontically Induced Tooth Movement"

_biomedicines, 2021, doi:10.3390/biomedicines9070796_

Round 1

Reviewer 1 Report

The manuscript authored by Dr. Kirschneck and colleagues aproaches a very interesting topic in current dental research.

I hope that my review and remarks will be useful for increasing the quality of their paper.

  1. Introduction - I would like that the authors will elaborate more on the RANKL/RANK system and its role in the periodontal disease etiopathology. In the same time I agree that OPG and RANKL/RANK play a crucial role in OTM but the statements should be more specific. I would recommend to include as well the role of MMPsas markers beside IL or TNF. Also a comparison between salivary markers vs. crevicular liquid ones might increase the quality of the paper.
  2. Materials and methods. Please elaborate the technical details regarding the particularities of the orthodontic appliances that you used in mice. This is extremely important in order to allow the replication of your experiment.
  3. Discussions section should be improved. I consider that is not sufficient.
  4. Conclusions must highlight more the practical aspect of the results and how controlling myeloid HIF1α may reduce periodontal hazards during orthodontic treatment. 

Author Response

The manuscript authored by Dr. Kirschneck and colleagues approaches a very interesting topic in current dental research.

Thank you very much for this assessment!

I hope that my review and remarks will be useful for increasing the quality of their paper.

  1. Introduction- I would like that the authors will elaborate more on the RANKL/RANK system and its role in the periodontal disease etiopathology. In the same time I agree that OPG and RANKL/RANK play a crucial role in OTM but the statements should be more specific.

We elaborated on the role of RANKL/OPG system in periodontal disease etiopathology, as suggested, and specified our statements about RANKL/OPG and OTM.

Revised text: RANKL/RANK signalling regulates osteoclast formation, activation and survival in normal bone remodelling, as well as in a variety of pathological conditions characterised by increased bone turnover [1], including periodontitis - recent systematic review and meta-analysis revealed that RANKL mRNA levels were significantly higher in gingival tissue as well as protein levels in gingival crevicular fluid (GCF) of individuals suffering from periodontitis [2]. Osteoprotegerin (OPG), which functions as decoy receptor of RANKL and can be released from osteoblasts as well as PDL fibroblasts, protects bone from excessive resorption by binding to RANKL and preventing its binding to the membrane-bound RANK receptor on osteoclast precursor cells and thus their differentiation and fusion to active bone-resorbing osteoclasts [1]. In this regard, not the absolute expression of RANKL is decisive for osteoclast activity, but rather the ratio of RANKL to OPG expression, both for orthodontic tooth movement [3], which required controlled osteoclastogenesis and osteoclast activity in the direction of tooth movement, as well as periodontitis [2,4]. A recent systematic review reports that (pre)clinical evidence regarding periapical as well as periodontal and lesions hints at an increase of the RANKL/OPG ratio as primary determinant of osteolytic activity in etiopathology of periodontitis, whereas a decreased RANKL/OPG ratio seems to be associated with inactive lesions[4]. Thus, the RANKL/OPG system plays a crucial role both in orthodontic tooth movement as well as in the etiopathology of periodontitis [4,5].

  1. Introduction I would recommend to include as well the role of MMPs as markers beside IL or TNF. Also a comparison between salivary markers vs. crevicular liquid ones might increase the quality of the paper.

We added a discussion about the role of MMPs and a comparison between salivary and GCF markers as suggested.

Revised text: Apart from the RANKL/OPG system, also a distinct remodelling of the extracellular matrix (ECM) of the PDL is required for OTM, which seems to be primarily controlled different matrix metalloproteinases (MMPs) and their tissue inhibitors (TIMPs) - both released by PDL fibroblasts upon force application [6]. These and various other molecules have been suggested as possible biomarkers in saliva or gingival crevicular fluid for monitoring or predicting OTM and possible side effects such as root resorptions. Allen et al. identified 20 OTM-related biomarkers in saliva in their systematic review [7] and Alhadlaq 34 OTM-related biomarkers in GCF [8] with some biomarkers seeming to occur both in saliva and GCF (RANKL/OPG, leptin) and others not (e.g. apoE, CLU, CRISP-3).

  1. Materials and methods. Please elaborate the technical details regarding the particularities of the orthodontic appliances that you used in mice. This is extremely important in order to allow the replication of your experiment.

We elaborated on the technical details regarding the orthodontic appliance as suggested, also referring to a recent publication reporting the used methodology in more detail.

Revised text: An experienced scientist inserted an elastic band (diameter 0.3 mm; Inwaria) between the first and second molars of the left upper jaw under anesthesia using two Mosquito clamps (straight, with teeth) and then shortened accordingly after pre-expansion with an orthodontic auxillary wire (Ø 0.08 mm), according to an established and validated protocol [9,10]. The diameter of 0.3 mm was the largest diameter insertable in the interdental space leading to a reciprocal movement of the first molar in anterior and the second molar in posterior direction upon relaxation of the compressed band [9].

  1. Discussions section should be improved. I consider that is not sufficient.

We improved the discussion section as suggested.

Revised text: The by tendency observed increase of Ptgs2 gene expression in Hif1αΔmyel mice during OTM, however, would be in accordance with previous research showing that Ptgs2 expression is regulated by Hif1α [11,12]. In macrophages, on the other hand, an uncoupling of compressive force and oxygen supply in vitro reduced HIF1α stabilisation without affecting Ptgs2 gene expression or prostaglandin E2 release [13].

Apart from controlling the response of cells to hypoxia and regulating immune responses, it is known that HIF1α is involved in skeletal development [14] and the control of bone resorption activity [15]. We detected impaired Rankl gene expression at the orthodontically treated jaw side in Hif1αΔmyel mice and by tendency also a reduced RANKL/OPG expression ratio, which could be due to impaired macrophage invasion or effects of myeloid Hif1α deletion on periodontal ligament fibroblasts. RANKL regulates the differentiation of osteoclast precursor cells to bone-resorbing osteoclasts, by binding to the RANK receptor on osteoclast precursor cells, initiating their differentiation and fusion to bone-resorbing osteoclasts [1,5]. Therefore, increased RANKL expression is associated with increased bone resorption [1]. This would explain the observed increase in orthodontic tooth movement velocity as well as reduction of bone density and by tendency of periodontal bone level. In our study, elevated Rankl gene expression was associated with increased Acp5 and by tendency Ctsk gene expression indicating enhanced osteoclast activity, as Acp5-coded TRAP (tartrate-resistant acid phosphatase) is a known marker for osteoclast activity [16] and Ctsk-coded Cathepsin K is an osteoclast-secreted protease able to cleave telopeptides and the triple helix of type I collagen fibers as well as matrix-metalloproteinase-9 (MMP-9) thus activating it [17], with both enzymes enabling the degradation of organic components of bone during bone resorption [3]. Studies on the role of HIF1α in bone development and remodelling demonstrated that HIF1α induction led to reduced osteoclastogenesis, but increased resorptive activity of mature osteoclasts [15]. In mice, HIF1α deletion is associated with a decrease in trabecular bone volume, as HIF1α is involved in osteoclast activation [18,19]. In contrast, overexpression of HIF1α was shown to reduce osteoblast differentiation, while osteoclastogenesis was elevated [20]. Initially, we detected no differences in alveolar bone density between wildtype and Hif1αΔmyel mice. After orthodontic treatment, however, mice lacking HIF1α in myeloid cells had a reduced bone density and accelerated tooth movement compared to wildtype animals indicating a bone-protective activity of myeloid HIF1α during orthodontic tooth movement, which is corroborated by our findings on increased osteoclast activity and RANKL expression under myeloid Hif1α deletion as well as by the at the same time not significantly altered Alp and Runx2 expression in Hif1αΔmyel mice during OTM, as Alp and Runx2 are marker genes for osteoblast activity [21].

  1. Conclusions must highlight more the practical aspect of the results and how controlling myeloid HIF1α may reduce periodontal hazards during orthodontic treatment. 

We highlighted the practical aspect of how controlling myeloid HIF1α may reduce periodontal hazards during orthodontic treatment.

Revised text: A targeted stabilization of myeloid HIF1α during orthodontic treatment, as was done experimentally via DMOG (dimethyloxallyl glycine) [11], might thus in the future be a possible therapeutic approach to reduce treatment-related periodontal hazards, which needs to be investigated in further studies.

Reviewer 2 Report

Dear Authors, please correct the following items: 

  1. chapter 2.3 title (page 4)
  2. explain what is ROI (page 4)
  3. write with italics ALP and Ptgs2 (page 5)

Question: please explain why did you choose seven days for the orthodontic treatment of the mice? How does it relate to human orthodontic treatment period?

Author Response

Dear Authors, please correct the following items: 

  1. chapter 2.3 title (page 4)

title corrected, as suggested

  1. explain what is ROI (page 4)

ROI = region of interest, we defined the abbreviation in the manuscript as suggested

  1. write with italics ALP and Ptgs2 (page 5)

done

Question: please explain why did you choose seven days for the orthodontic treatment of the mice? How does it relate to human orthodontic treatment period?

This time period and methodology used to induce experimental tooth movement was previously established and published to be suitable for experimental studies on orthodontic tooth movement in mice [9]. As metabolism in mice is much faster than in man, 7 days of tooth movements corresponds to a much larger time period in man, albeit it is not possible to determine the corresponding time period in man.
